# *Line-Storm* Ludic System: An Interactive Augmented Stylus and Writing Pad for Creative Soundscape

Category: Research

Paper Type: system

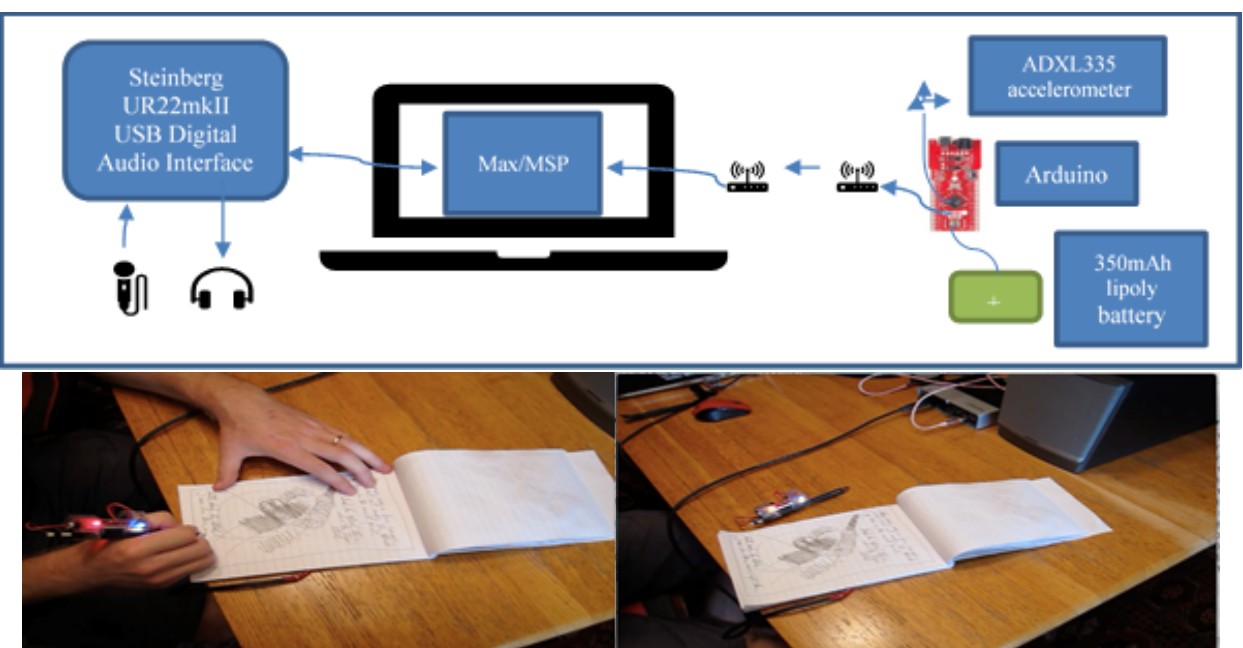

Fig. 1. a (above): *Line-Storm*s System Diagram. Fig. 1b-c (below) Line storm stylus drawing and sample of creative content

**Abstract**—We present *Line-Storm*, an interactive computer system for creative performance. The context we investigated was writing on paper using *Line-Storm*. We used self-report questionnaires as part of research involving human participants, to evaluate *Line-Storm*. *Line-Storm* consisted of a writing stylus and writing pad, augmented with electronics. The writing pad was connected to a contact microphone, and the writing stylus had a small micro-controller board and peripherals attached to it. The signals from these electronic augmentations were fed into the audio-synthesis environment Max/MSP to produce an interactive soundscape. We attempted to discover whether *Line-Storm* enhanced a self-reported sense of being present and engaged during a writing task, and we compared *Line-Storm* to a non-interactive control condition. After performing statistical analysis in SPSS, we were unable to support our research hypothesis, that presence and engagement were enhanced by *Line-Storm*. Participants reported they were, on average, no more present and engaged during the experimental condition than during the control condition. As creativity is subtle, and varies with person, time, context, space and so many other factors, this result was somewhat expected by us. A statistically significant result of our study is that some participants responded to *Line-Storm* more positively than others. These Preservers of *Line-Storm* were a group, distinct from other participants, who reported greater presence and engagement and who wrote more words with *Line-Storm* and during the control condition. We discuss the results of our research and place *Line-Storm* in an artistic-technological context, drawing upon writings by Martin Heidegger when considering the nature of *Line-Storm*. Future work includes modifying interactive components, improving aesthetics and using more miniaturized electronics, experimenting with a drawing task instead of a writing task, and collaborating with a composer of electronic music to make a more interesting, immersive, and engaging interactive soundscape for writing or drawing performance.

**Index Terms**—Ludic System, Creativity, Interactive Soundscape

---

## 1 INTRODUCTION

Our philosophy is that people have become frugal regarding "joy"! How we all are becoming increasingly suspicious of all joy! The desire for joy already calls itself a "need to recuperate" and is beginning to be ashamed of itself. –Nietzsche [51]

Tod Machover [47] has emphasized the need to augment existing, traditional musical instruments while ensuring these augmentations act as *stimuli* to the creative process, not simply as additional features.

One focus of this paper is to find a way to enhance human creativity. Another is to observe the emergence of the work when the system is used. A third, is our attempt to make something that is fun to use. We have conceived, designed, constructed, evaluated, our system called *Line-Storm*[1], attempting to enhance a sense of both presence and engagement in the user. Only through performance with *Line-Storm*, does *Line-Storm* come into being.

---

[1]We chose the name *Line-Storm* after a favorite Robert Frost poem, "A *Line-Storm* Song."

The method of experience sampling–interrupting a person as they go through their daily activities and asking questions about their experience–has been used to find that when peoples minds are wandering, they are less happy [43]. "Be Here Now," a mantra popularized in the United States by, for example, Dr. Richard Alpert [18], who became Baba Ram Dass. This mantra now occurs in a leading business publication urging middle managers everywhere to "be present" to be a "great leader" [35] and presumably to reap the rewards of "success." Even the LSD experimentation Dass describes in Be Here Now, carried out on a small, socially acceptable scale in Silicon Valley, where tech workers "microdose" themselves with LSD, to enhance their creativity and improve interpersonal interactions [45]. Some esoteric practices leading to creative work may conjure images of the lone painter or poet, or of a sculptor in her studio. It is not only Silicon Valley technocrats, scrambling for millions and billions of dollars, who might benefit from enhancing human creativity.

> Even now one is ashamed of resting (equated to waste of time in our mind), and prolonged reflection almost gives people a bad conscience. One thinks with a watch in ones hand, while eating meals, and reading the latest news of the stock market; we live today not to miss out on anything. –Nietzsche [51]

Note that Nietzsche was writing well over 100 years before "FOMO," or "fear of missing out," became an expression related to early 21st-century smartphone users. Our point is that we recognize that there are different meanings to the phrase creative work. For example, billionaires and poets are not endorsing the same thing when both use the word "creative" or the word "work," though both may praise "creative work."

Some decry the extreme measures taken by LSD trippers in the 1960s [45], and want to turn the drug into an effective money-making tool. An irony is that creative work translates into fortunes undreamt of by poets such as Robert Frost. There is a story in which Joseph Heller, author of the novel *Catch-22*, when told of an investment banker who had made more money last year than he might ever to be expected to make from the novel, replied that he had something the investment banker would never have: *enough*. So, we argue that it is possible that what was good for Heller, in the anecdote, would probably not have been good for the investment banker, even when the concept of creative work is broadened to include both their endeavors. Enhancing one type of creative work may not enhance the other. The ecstasy of the composer remarked upon by Csikszentmihalyi [15] or of the novelist, may not be found in the same way the "A-ha!" of the software developer is found.

Our work involving *Line-Storm* has been an attempt to provide a ludic system for use by the creative worker. Gaver [24] defines a ludic system as one that is used for its own sake, and not for some other end. By attempting to increase a users sense of presence and engagement–their being here now–our hope is to provide an immersive environment in which to do creative work with a writing stylus such as the mechanical pencil we chose to use. Taskscape is a complex term from Ingold's "The Temporality of the Landscape" [38], which we will refer to later, when speaking of the new possibilities of a task that *Line-Storm* exposes, as *affordances* in Gibson's sense of the term [22]. One of our committee members, a professor of music, suggested that our work involves the taskscape of the creative worker, working with a writing stylus and paper. This taskscape includes the place, people, and objects surrounding the creative worker doing creative work. The taskscape is social [38]. The experience of the user of our system, and of the research participants who gave of their time to be a part of this thesis, is a social experience, and the writing tasks they performed are tasks that fit into "an array of activities"–which include the writing of this sentence [38]. We do not know–as above, because too little work has been done in this area–whether the taskscape of a user of *Line-Storm* is altered in ways more conducive to writing poetry than to the drafting of microprocessor plans, for example, or vice versa.

Rather than devise a completely new tool, we have chosen to augment an otherwise ordinary mechanical pencil[2]. Perhaps by looking

---

[2]We could have similarly augmented a paintbrush or a pen, though the

away from our goal, creative enhancement–as we must when looking at faint night-sky objects with the naked eye (Springob, 2015)–and making the use of the system the primary activity, and the work done with it a secondary activity, we think we will find ourselves progressing in that direction, whereas a direct approach would not have succeeded. By giving a chance for play, we have hoped our system, *Line-Storm*, serves as stimulant and facilitator "to the creative process itself," as Machover [47] advises.

## 2 RELATED WORK

*Line-Storm* is not digital art as such. Its products are physical objects and phenomena. They are (analog) drawings or writings. It produces sounds, which are–though digitally mediated–analog sounds. The computer is, in *Line-Storm*, an intermediary and a facilitator, with a visual arts component and sound, satisfying criteria for Demers' [8] second sub-genre of sound art. *Line-Storm* amplifies and augments the sonic aspects. The sounds made, while writing or drawing, are captured using a contact microphone and are played through headphones. Sounds of natural phenomena–the sounds of a thunderstorm–augment the writing or drawing experience. These sounds are recorded-analog yet are digitally mediated (Steven Jesse Bernstein (1992)).

### 2.0.1 *Line-Storm* as Performance

Performance with *Line-Storm* can involve performance in art, in technology, and/or in play [58]. A performer using *Line-Storm* may be using it for different reasons, including for the fun of using it, to write a letter to a friend, to write down a cooking recipe, to write poetry, to draw, because it is a curious thing one wants to understand, or for other reasons. A performance occurs "as action, interaction, and relation" [58]. *Line-Storm* is an interactive system, where the performer's actions cause sounds to occur, which may influence subsequent actions. The sounds can be controlled to some degree, by the performer. The drawing or writing produced during performance is one product of the performance. The sounds, which can be recorded and played back, are another product. The audience of the performance may be the performer alone, or a person or persons presented with one or more products of the performance, the written or drawn product or the sound produced. *Line-Storm* is a way of "honoring the ordinary" in Schechner's [58] words.

Schechner [59] refers to "Nietzschean" play, making specific reference to a dice roll. This is contrasted with playing a game with rules that are agreed upon by everyone before play starts [24,59]. Play is a way of introducing flow into ones life [59] and has an organic quality. Deleuze [20] has discussed Nietzsches approach to the dice roll, in *Nietzsche and Philosophy*. Sounds which we added to *Line-Storm* included those of thunderstorms, which have organic qualities similar to movement of air through a room, pushed by a ceiling fan, or sounds from nearby birds. Thunderstorms followed by quiet rain can be medicative to some as well. We wanted to use analog (thunderstorm) sounds for our analog ludic system. Gamification is contrary to play and playfulness [24] and renders "personal" relationships impersonal. Bohme [5] has commented upon the superfluity of personal relationships to the normal functioning of society. The digital medium is one of permanence and impermanence. Privacy becomes a concern in the digital realm, in ways not found in the analog realm. Talk of a "right to be forgotten" has occurred surrounding the at-times oppressive permanence of the digital medium. *Line-Storm* as a medium provides the privacy familiar from the analog world. Letters can be (and have been) intercepted, shared unexpectedly, and so on, but personal letters written on paper are not automatically added to databases, compiled on users of internet services. One motivation of *Line-Storm* is that it could preserve the practice of the handwritten letter.

### 2.0.2 Previous and Related Work

> No human tools are more beautifully designed for their purpose than traditional musical instruments, which no col-

---

paintbrush would have required a different approach. We depend in part on the sounds made by the user's touching of the writing pad, and we cannot expect a paintbrush to make the same level of sound made by a pencil lead.

lection of buttons, wires, and sensors can replace. That's why it is important that technology be used to augment, not replace, existing instruments. –Tod Machover [49]

Previous work, that investigated augmenting a writing stylus with electronic or computer systems, includes MusicGrip [26], a pressure-sensor-controlled system in which a writing stylus was used to control analog synthesizers. Musc Grip used a one-to-one correspondence between sensor input and synthesizer output. Shichinohe *et al.* [60] used a camera system to implement an augmented-reality system to aid in the instruction of calligraphic writing (Shichinohe, Yamable, Iwata, & Nakjima, 2011). Their system monitored brush position and body posture, providing both ambient (color) feedback and verbal feedback. Part of a performance–the *Brain Opera*–the Digital Baton was a wireless baton, augmented with sensors, used as a New Interface for Musical Expression (NIME) [52]. The baton carried an infrared LED at its tip, pressure-sensitive resistors that were controlled by the performer's fingers gripping the baton, and three +/-5g accelerometers. These inputs were mapped to musical parameters. The Digital Baton was a wired NIME, but the authors did discuss what could be done to make it wireless.

Much work relevant to ours has been done by Tod Machover, whose research group at MIT's Media Lab developed the technology behind Guitar Hero [34]. His work with *hyperinstruments* (electronic augmentations of traditional music instruments) [49]) and [47] are very interesting. Machover's Hyperstring Trilogy [48] was composed and performed using hyperinstruments–hypercello, hyperviolin, and hyperviola–which were traditional classical instruments augmented with sensors. The entire performance was itself augmented using computer programs that generated accompanying musical sounds. Machover's philosophy of augmenting, and not replacing, traditional tools, is one we have followed in our work [49]. LiveScribe (http://www.livescribe.com), which has produced a wireless pen with handwriting recognition, no longer develops the electronic writing pen it once did, so we did not involve the company's work in our work.

Work involving the augmentation of objects other than writing utensils or musical instruments includes the Sonic City system [25], in which the urban environment served as the interface. Sonic City equipped an urban pedestrian with a music-generating computer that was "aware" of the way the user traversed the streets and sidewalks of the city [25]. The Bluetooth Radio Ball Interface (BRBI) [64] augmented a sport ball with sensors, providing sound and music capabilities (mediated by a computer and a Bluetooth radio connection). The Urban Musical Game [54] was another augmentation project involving a sport ball and sound/music generation based on the ball's motions; video of use of Urban Musical Game have been made available on Vimeo (https://vimeo.com/26413625) and (https://vimeo.com/22120867). Measurement of writing motions helps diagnose people suffering from obsessive-compulsive disorder (Mavrogiorgou, et al., 2001). Handwriting and cell-phone texting have been compared as therapies for Brocas aphasia, with handwriting emerging as the more effective treatment [4]. Embodied cognition models have been used to investigate neural relationships with character writing, copying, and recognition [41]. Preschool children have taken part in fMRI experiments, which demonstrate the importance of "learning-by-doing" approaches to literacy learning, with kinesthetic activity working in tandem with cognition [40]. Existential phenomenology has informed thought regarding the teaching of personal writing, without any technological involvement [39]. Recent work by Kiefer [42] has found neuropsychological evidence for benefits from writing by hand, as opposed to writing using a computer keyboard, including improved learning of reading and writing skills in young children. Morphy and Graham [50] argue students more generally, appear to write better when using word processors than when composing by hand, considering the composition tools (spell check, grammar check, etc.) available in modern word processing software. Al-Ghabra [1] focused on the importance of handwriting for the development of composition skills in college students. Earlier work by Collier and Werier [7] found no difference between high-level characteristics of textual production in proficient adult writers who composed either by hand or while using a word processor.

## 2.1 Development of Work

An early impetus for our work was a desire to encourage handwritten letter writing, to provide a system that might encourage a person to increase or maintain their stamp-and-envelope transmission of handwritten letters. Thoreau [63] decried some forms of letter writing, writing that, "The penny-post is, commonly, an institution through which you seriously offer a man that penny for his thoughts which is so often safely offered in jest"; from a different viewpoint, writing letters has been a way for families to stay connected through the generations and has functioned alongside newer media [53]. Twentieth-century German philosopher Martin Heidegger commented, in his Parmenides, upon handwriting, declaring its superiority over use of a typewriter [33]. Philosopher of technology Don Ihde [37] faulted Heidegger for Heidegger's comparison. Philosopher Jacques Derrida [21] also faulted Heidegger, for implying, while emphasizing the importance of "the hand" for humanity, that human beings only have one hand. A typewriter does not offer the affordance [22] of being easily carried up a mountain–although Nietzsche owned a portable typewriter [3]. Likewise, the poem title of a friend, "Notebooks," [46] would read differently if it had to do, not with notebooks, but with some digital note-taking contrivance such as Google's Keep app (http://keep.google.com). Ihde reminds us of the non-transparency of electronic and digital communications media such as the telephone [36], and here, with Nietzsche's typewriter and Gregory Lawless's poem, we see some effects of medium, in practice (typewriter) and in discourse (poem title). Ihde does not want to go with Heidegger, to declare there is not just a difference between media but a hierarchy of values. These dismissals, by Ihde and Derrida, of Heidegger's criticism of the typewriter and his preference for handwriting, seem to pretend to have the benefit of hindsight. Ihde and Derrida do not convince us, because they, like the king Thamus, do not look enough at what might support Heidegger's position, but only criticize it–their seeming invincibility coming from their objections to Heidegger being voiced decades after his death in 1976 [44].

Heidegger [30] decries what he sees as hastiness in the face of a technologically facilitated information glut: "[N]owadays we take in everything in the quickest and cheapest way, only to forget it just as quickly, instantly." Both Heidegger [30] in his "Memorial Address," and Jacques Ellul [23] in *The Technological Society*, declare technology to have become "autonomous" (in Ellul's phrasing), saying its progression could not be stopped, even if human beings *wanted* to stop it. "These [technological] forces have moved long since beyond his will and have outgrown his capacity for decision," Heidegger says, regarding our relationship to technology and "calculative thinking" [30]. Our thinking here is that technology creates more options, including the option to not use it; non-users of a technology have been considered by Satchell and Dourish [56]. These are core questions we consider as we discuss *Line-Storm*. Should we augment human capability, or should we replace it with a technological contrivance? As discussed above, we have followed Machover in choosing to augment human creative capability, using *Line-Storm*.

### 2.1.1 Creativity and *Line-Storm*

For work done by Csikszentmihalyi, in *Creativity: Flow and the Psychology of Discovery and Invention* [13], ninety-one persons were interviewed who were deemed to have made significant contributions to their fields [11]. Many others, who excluded themselves from his study, were skeptical of studying creativity or of participation in the study as being worthy of their time, and some insisted they were too busy being creative to stop and talk about it [14]. A direct approach to enhancing creativity, Csikszentmihalyi [14] writes, is less effective than are attempts to place the creative worker in a favorable environment; but beautiful surroundings are not what he means. The creative worker creates an environment conducive to creative thought and work, despite otherwise unfavorable surroundings; "they manage to give their surroundings a personal pattern" [16]. On the other hand, he denies

there is proof that a person needs "delightful" surroundings to engage in creative work [12].

## 2.2 Conceiving Creative Work with *Line-Storm*

There are emotional aspects as well–creativity is not only cognition, taken as reasoning–and attention is the finite resource that allows the creative person to find problems to solve, that have gone unrecognized until they have found them [16]. Our work attempts to alter the state of the creative worker, short of accomplishing shamanistic technique or administering psychedelics . Csikszentmihalyi makes a similar claim for the creative worker. The creative worker has their attention focused in areas outside the "status quo" (Csikszentmihalyi, Motivation and Creativity: Towards a Synthesis of Structural and Energistic Approaches to Cognition, 2014). Creativity is lauded widely yet creativity works for good and bad. Cropley [9, 10] wrote that a computer hacker who circumvents security measures to steal money, has exhibited creativity no less than a symphonic composer imagining a new melodic line. Sternberg [62], writing of what is known about creativity, iterates two points: (1) creativity is mostly "domain-specific," and (2) it is partly independent of measured intelligence quotient (IQ).

## 2.3 Creativity as Play

Much ink, including that of Thoreau [63], has been spilled comparing creativity to play. Play does not need the context of a game, to be play. Play may be contrasted with the world of production and work. In attempting to provide an immersive experience conducive to the presence and engagement of the creative worker, we recognize the worker–even a solitary worker writing alone in a micro-environment they have fashioned to their needs–as engaging in a performance. The act of sitting down to write a poem might be viewed by some poets as requiring a "smooth" execution; but others will likely view it as Schechner [58] has described the "actual" and the roughness of the performance of writing the poem as "the genuine meeting between performer and problem"; having a sense of presence and engagement is a desirable state.

### 2.3.1 Concerns Regarding the Need for *Line-Storm*

*Line-Storm* is interactive. It augments an ordinary pencil and an ordinary pad of paper, adding new interaction possibilities, new affordances [22]. It responds to the person engaged in using it–and it is immersive. The headphones may make it "easy to forget the outside world," allowing the user to "concentrate completely" on the writing task (Csikszentmihalyi, Creative Surroundings, 1996).

*Line-Storm* is an attempt at providing creative workers with a new tool. Citing Edward Tenner, Runco [55] cautions that tools do not have to be poorly made or poorly designed or have "an undesirable feature, to cause problems" involving either the creative worker or others. Combining technology with art and art works does not, in itself, enhance creativity [55].

Heidegger [32] inquired into whether the common conception of technology is neutral and independent of its uses, was supportable, and found the essence of technology to be a view of the world and all that is in it, as resource to be put to use, or standing-reserve. Han, in *In the Swarm* [27] and *Psychopolitics* [28], and Bohme [5], in *Invasive Technification*, question the place of technologies in our lives, and the role of the associated, technological, perspective in dominating other forms of life. Lucas observed of the world depicted in his 1971 film, *THX 1138*, that "nobody was having any fun, but no one was unhappy" [19]. We have made a new piece of technology that is based on fun. We tread softly when we attempt to bring new technologies into the practice of writing or drawing by hand with pencil and paper.

### 2.3.2 Enhancing Creativity through Inefficiency

Dan Ariely [2] posed a thought experiment, in his 2010 book *The Upside of Irrationality*, in which the reader was asked to imagine how a large cash reward might motivate them to greater creativity. After he considered the experimental evidence, he wrote that money is a poor motivator to creative production [2]. According to Ariely [2], it is not clear how much of our "mental activity" is under our "direct control," especially when we are working under pressure. Our system might prove more difficult to use than ordinary pencil and paper, and this is not in itself a problem for us because creativity is different than usability. In addition creativity may not be fully mechanizable. Creativity is not only randomness or sheer novelty; it requires filtering by an intelligence [57]. Counterintuitive incentivisation may be called for when attempting to stimulate creativity. Making a task more difficult through the use of unusual tools, may stimulate creative production. Changing the affordances [22] of a once-familiar taskscape [38], may be key to inducing creative thought, making one see a thing or activity in a new way.

## 3 IMPLEMENTATION

We present the details of our implementation of *Line-Storm*.

## 3.1 Development Environment: Max/MSP

We implemented the software interface and sound-synthesis engine of *Line-Storm* using Max/MSP, Version 7.2.3, 64-bit edition. Max/MSP is the mature, commercial successor to Miller Puckettes Pd (aka Pure Data) (https://puredata.info/downloads/pure-data), a free and open-source project. Like its predecessor, Max/MSP is a graphical programming environment. Objects in the Max/MSP GUI windows can be interconnected and otherwise manipulated inside patchers (graphical representations of program files in Max/MSP). A Max/MSP program or patcher appears generally as one or more objects connected by patch cords joining inputs and outputs. Inputs and outputs can be: symbolic, that is, textual; numeric; or signals running at an audio rate, typically 44.1 kHz. Max/MSP allows us to interface with a wider range of electronic devices, and we have elected to use the serial-port communication capabilities of Max/MSP to bring our sensor data into the Max/MSP application, as a control signal, through a serial port on our laptop computer.

Max/MSP has further advantages over some other music-synthesis DAWs such as FM8; Max/MSP is programmable, and it is well-documented.

## 3.2 Sensor-Fob Construction

The sensor-fob, shown above, in Figure 17, comprises multiple PCB circuit-boards, powered by a lithium-polymer battery, and five solid-core, insulated copper wires soldered between two of the PCB circuit-boards.

The primary board is an Arduino Fio v3 microcontroller board. This type of Arduino board includes a socket into which an XBee radio transceiver module can be inserted (see Figure 21, below). The Fio v3 can control the inserted XBee radio transceiver module. We have a Digi International XBee radio transceiver module (type S1) inserted into the socket of our Fio v3. The Fio v3 has multiple GPIO/ADC (general purpose input/output or analog-to-digital converter) pins, three of which we have soldered wires to. These three wires are soldered at their ends, to an Adafruit ADXL335 3-axis accelerometer, to its three, analog output-signal pins. Two more pins, and two more wires, connect Vcc and GND on the Fio v3 and ADXL335. The wires are rigid; they both connect the boards and hold them in constant positions relative to each other, in a fixed orientation. See Figure 22, below, which shows the solder connections between the Fio v3 and the ADXL335.

We chose to orient the boards parallel to each other, the top or front face of the Fio v3 facing the bottom or back face of the ADXL335. The axes of the ADXL335 3-axis accelerometer are oriented along the lengths, widths, and surface normals of both parallel boards. The x-axis of the sensor is oriented along the length of the Fio v3; the y-axis along the width, and the z-axis along the surface normal. The adhesive-tape connection of the sensor-fob to the stylus allows quick changes in the relative orientations of the stylus and sensor. When carrying out experiments involving human subjects, we maintained a constant relative orientation of sensor-fob to stylus, with the XBee-radio end of the sensor-fob pointing in the direction of the eraser on the pencil-stylus, as shown in Figure 16, above.

### 3.2.1 Accelerometer

We used an Adafruit ADXL335 3-axis accelerometer mounted to a breakout board (see Figure 20, which also shows solder points). The ADXL335 has an acceleration-measurement range of +/-3g, where g is the average acceleration due to gravity at Earths surface. While there was a similar model, available from Adafruit, which had a +/-5g range, we found that the ADXL335 +/-3g measurement range was sufficient for our system. Moving the sensor-fob, with its attached ADXL335 unit, in as violent a manner as we were able to do while holding it with a hand, we sometimes reached minimum and maximum sensor-output values, but not always; reaching these values was difficult. Lesser motions were well within the +/-3g range, giving sensor output values below the approximately 1000 maximum and above the approximately 0 (zero) minimum. Sensor output values, raw from the GPIO/ADC pins, range from 0 to approximately 1000, with a center value of approximately 500. This range is compatible with an 8-bit ADC, which the Fio v3 uses. Values below about 500 indicate negative accelerations relative to the corresponding sensor axis, while values above 500 indicate positive accelerations relative to the corresponding sensor axis. More programming and other details can be found in [FirstSecondAuthor].

## 4 EVALUATION

We present the details of our experimental evaluation of *Line-Storm*, which involved research involving human participants.

### 4.1 IRB Study: Our Participants

Our study involved participation by thirteen persons, but data for one of these participants was discarded, leaving twelve participants with valid data. The participant whose data we discarded had filled out the questionnaires by circling the value 3, in all cases but one, on both *Line-Storm* and control-condition questionnaires, strongly suggesting they did not follow directions or complete the forms in good faith. We had roughly half female and half male, including one who chose not to self-identify. Participant ages are ranged from 18 years to 34 years. We ran our experiments at two sites. We began advertising near the end of the Spring semester, and it seems likely many students were interested in obtaining extra credit at that time. In three days, we were able to run ten participants.

## 5 DATA ANALYSIS AND RESULTS

We present and analyze the results of our research study.

### 5.1 Summary of Results

We were unable to support our research hypothesis, that participants' sense of presence and engagement would be greater during the experimental, interactive condition than during the control, non-interactive condition. Because creativity and engagement is somewhat fleeing and vary from one person to another, this was not so unexpected result. Using our abbreviation for control and experimental questionnaires' first item, measuring self-reported level of presence and engagement–PANDE for "present and engaged," a suffix of 0 (zero) referring to the control condition questionnaire item and a suffix of a 1 referring to the experimental condition questionnaire item–we found we could not reject the null hypothesis, PANDE0 = PANDE1. We compared mean self-reported level, of being "present and engaged" during the experimental and control conditions, using two statistical tests. We performed a paired-samples test using the Student's t distribution, suitable for small sample sizes from normal populations, and we performed a paired-samples test using the nonparametric, distribution-free Wilcoxon Signed Rank Test, suitable for symmetrical distributions of small samples.

We performed these two statistical tests for all paired variables, that is, for every matched item on the control and experimental questionnaires. We did find a statistically significant difference between experimental and control conditions, for two questionnaire items, abbreviated NAT (for naturalness of interaction with the system) and ADJEXP (for adjustment to the system experience). Both these items were rated lower for the experimental condition than for the control

condition, indicating participants found the experimental system interactions unnatural and had difficulty adjusting to using it, compared to the control condition.

We performed Pearson correlations, and found several statistically significant correlations, discussed below. For example, those participants who reported they lost track of time during the experimental condition also tended to write more during the experimental condition. There was a non-significant correlation between losing track of time and word count during the control condition. One participant circled 3 for all questionnaire items on both the control and experimental questionnaires, except for one item for which this participant circled 3. We excluded this participant from our data analysis for this reason. They did not appear to follow the instructions for the experiment. When examining word counts (WC0 and WC1), we excluded two participants who drew pictures instead of writing, leaving a sample size of ten.

We performed K-means clustering classification, discussed below. A group of participants responded differently to the experimental condition than did the rest of the participants. There was also a group who responded differently to the control condition than did the rest of the participants and we have termed these *Preservers of Line-Storm*.

### 5.2 IBM SPSS Statistics Version 25

To perform our statistical analyses, used IBM's SPSS Statistics, Version 25 (https://www.ibm.com/products/spss-statistics), because it is an industry standard statistics processing application. Many tutorials are available online, in textual/graphical and video formats. SPSS also has links to user forums and documentation integrated into the application.

### 5.3 Summary of Correlations

- There were strong, significant ($p<0.01$) correlations between the initial, baseline level of a sense of presence and engagement and response items 4 (NAT) and 7 (ADJEXP), for both control and experimental conditions.

- A sense of presence and engagement correlated strongly and significantly ($p<0.01$) with adjustment to the "control devices" (augmented stylus, augmented writing pad) (ADJCTL) for both control and experimental conditions.

- There were strong, significant correlations between a sense of the naturalness of interactions with the system and baseline sense of presence and engagement, ease of adjustment to the system experience, and ease of adjustment to the control devices (ADJCTL), for both control and experimental conditions.

- There was a group of participants who responded more favorably to the experimental condition than the rest of the participants (analysis performed using K-means clustering tests). *This is significant result for our experiments.*

- Those who wrote more in the control condition wrote more in the experimental condition. *This is also significant result for our experiments.*

- The more participants lost track of time, in the experimental condition, the more they wrote–or vice versa. *This is significant result for our experiments.*

- We found correlations between a sense of presence and engagement during the experimental condition (PANDE1), and the degree to which a participant lost track of time while using the system during the experimental condition. *This is significant result for our experiment, and we call these participants as the* **Preservers of Line-Storm** .

## 6 DISCUSSION

We discuss our experimental results.

## 6.1 Comparing Control and Experimental Conditions

We found that participants found their interactions with the system more natural during the control condition, and less natural during the experimental condition. Participants adjusted to the system experience more quickly during the control condition than they did during the experimental condition.

## 6.2 Comparisons Using Wilcoxon Signed Rank Test

Our findings indicate that there appears to have been a significant group of participants, roughly half the participants, the Preservers of *Line-Storm* (see below, What *Line-Storm* Is: Equipment, Art Work, and Preservation), who became immersed during the experimental condition. These participants tended to write more during the control and experimental conditions, they tended to experience the sound components of the system (control and experimental) in a way that led to their reporting less prominence of the visual aspects of the system, and they tended to lose track of time during the experimental condition.

## 6.3 Possible Natural Fluctuation of Presence and Engagement

It seems likely that attention fluctuates over time, and the mind naturally wanders and returns. Future work would include investigation of the ways such natural fluctuations in attention would be relevant to our work. Considering the ordering of questionnaire completion was nearly always the same (Demographic, Experimental, Control), natural fluctuations in attention (and presence and engagement) may help to explain our results. Considering how we might have been wearing out our participants, by making demands upon their attentional resources, future work might be done that minimized attentional fatigue.

## 6.4 What *Line-Storm* Is: Equipment, Art Work, and Preservation

When interpreting our results, we are interpreting the interpretations of our study participants. Their interpretations played a role in the coming-into-being of *Line-Storm*. Martin Heidegger wrote that, "Just as a work cannot be without being created, but is essentially in need of creators, so what is created cannot itself come into being without those who preserve it" [31]. With reference to Heideggers essay [31], "The Origin of the Work of Art," we see two ways *Line-Storm* can be approached. First, as equipment, *Line-Storm* is a tool we have made for a purpose. As equipment, *Line-Storm* has a thingly character and an equipmental character. As a thing, *Line-Storm* exists as an object that can be encountered in the world, like a rock. As equipment, *Line-Storm* is experienced as part of a matrix of all equipment. It exists in a context of equipment, purposes, and activities. As ready-to-hand, it withdraws [29], becomes transparent, and the person using it is engrossed in the work. As present-at-hand, *Line-Storm* sticks out as a thing [29], instead of becoming transparent and allowing the user to become engrossed in the work.

As art work, *Line-Storm* is created, not only made. It has a thingly character and a workly character. Preservation "does not degrade it to the role of a stimulator of [mere lived experience]" [31]. It is not a tool in this case, where it would be "released beyond itself, to be used up in usefulness" [31]. As art work, *Line-Storm* is bringing forth of the work that there lies this offering that it be [31]. These two modes of being that *Line-Storm* permits, refer to two ways of being with *Line-Storm*: as technician or as performer. For a technician, *Line-Storm* is equipment. Equipment serves a purpose and is not an end but a means. As Schopenhauer declared, equipment and "all other human works," that are not art works, "exist only for the maintenance and relief of our existence"; art works "exist for their own sake" [17]. When we evaluated *Line-Storm* in terms of its capacity for leading to a possible increase in self-reported presence and engagement, we treated it as equipment. Yet some participants, while using *Line-Storm*, treated it not as equipment but as art work. Hence, we will refer to the group of participants who gave higher ratings to *Line-Storm*, and who wrote more while using it, as the Preservers of *Line-Storm*. The Preservers let *Line-Storm* be what it is. Without the Preservers, *Line-Storm* "cannot itself come into being" [31]. Preservers here simply means to us are those participants who seem to like *Line-Storm* and used it enough so that to be creative in their own mind.

## 6.5 Finding the Affordances of *Line-Storm* in Preservation

*Line-Storm* permits itself to be used in performance. A performance with *Line-Storm* could be understood to point out the overlapping of sensory or perceptual modes commonly thought of as separate. Seeing, hearing, moving, and proprioception involve cross-modal transfer [61]. The sound and visual aspects overlap more strongly in *Line-Storm* than in ordinary writing or drawing, because of the amplification of what had been quiet sounds, i.e. the sound made by stylus on the paper which was amplified and merged with other sounds, such as thunderstorm.

We perceive an object or art work as affordances, the tripartite interrelation of environment, organism, and activity [22]. Because of cross-modal transfer, sound, visual, and tactile affordances play together in *Line-Storm*. When we look, we see what we can do–although, as with a wooden sculpture-puzzle, we may not immediately see all affordances. To solve the puzzle is to discover hidden affordances. *Line-Storm* makes affordances prominent, in the writing stylus and writing pad, that may not have been apparent: their sound-producing capabilities, which can be used in a performance. Preservers of *Line-Storm* find its affordances. This is a kind of knowing; the work of preserving a work is know-how [31]. A performance with *Line-Storm*, as preservation, is a knowing and a realization of the affordances of *Line-Storm*, without which it cannot be *Line-Storm*. Preservation is the fulfillment of the work that is *Line-Storm*. Both preservers and we, as creators of *Line-Storm*, belong essentially to its creation [31].

## 6.6 Robotany and *Line-Storm*

In 2006, a living Japanese maple tree was augmented with nitinol wires and optical and audio sensors. The tree moved its branches, using the nitinol "muscle" wires, in response to the presence of people detected by its sensors. Coffin [6] discussed this art work in the context of Heideggerian phenomenology, and discussed two different implementations of the tree totem, Breeze. In one, Breeze was constructed using a live Japanese maple. The flexible branches of the maple, and the bushy shape of the tree, hid the mechanical components from view, and attendees of the festival, where Breeze was exhibited, tended to interact freely with the tree and differently than during the second exhibition. At the second exhibition, a mountain laurel was used, whose stiffer limbs and more open shape put the mechanical components of the installation on display. When the mechanical components were hidden from view, during the first exhibit, interactions took place with people treating the tree as ready-to-hand. The mechanical components of Breeze withdrew, became transparent, and the people at the first exhibit interacted with Breeze as an interactive art work. When a tool is present-at-hand, the tool exists differently for the person. A broken hammer is present-at-hand as an object, not ready-to-hand as a useful tool. A broken hammer does not allow the person to engage in the work but is open for inspection. During the second exhibit, with the mechanical components poorly hidden by the tall, open shape of the mountain laurel, attendees at the exhibit tended to comment on the engineering of Breeze instead of interacting with it freely as had the attendees during the first exhibit. This is quite interesting for *our* implementation as well.

*Line-Storm* appears to have existed differently for different research study participants. We propose that those who seemed to enjoy using *Line-Storm* engaged with it as ready-to-hand, while those who did not appear to enjoy using *Line-Storm* engaged with it as present-at-hand. For the former, *Line-Storm* became transparent and withdrew itself, allowing the participants to write or draw as well as they would have with an ordinary pencil and paper. For the latter, *Line-Storm* obtruded and interfered with the writing task. Those participants for whom *Line-Storm* was present-at-hand were not able to engage with the work but were distracted by the strange contraption, the mechanical and interactive aural properties of the device. We propose that Csikszentmihalyi's concept of flow implies engagement with tools and tasks in a ready-to-hand mode. A tool being present-at-hand is indicative of the absence of the flow state.

The peasant woman wears her shoes in the field. Only here are they what they are. They are all the more genuinely so, the less the peasant woman thinks about the shoes while she is at work, or looks at them at all, or is even aware of them. She stands and walks in them. That is how shoes actually serve. It is in this process of the use of equipment that we must actually encounter the character of equipment. [31]

*Line-Storm* is equipment, when a participant is using it as equipment [29]. Its equipmental character manifests when the user is engaged with the work and not distracted by the system's appearance or feel. Its "thingly" character is manifest [29] for a participant who is not having fun using it, is not in a flow state, and is not present and engaged while using it. We suggest that these latter participants did not experience *Line-Storm*. One component of *Line-Storm* is its interactivity. A certain type of actions and intentions are required on the part of the participant for *Line-Storm* to be what it is. Coffin wrote of Breeze, and of interactive systems more generally, that interactions with them may be "effortless, unscripted, emergent, and engaged" if the mapping of responses is well done with respect to our "meaning-making sensibilities" [6]. Our goal that *Line-Storm* would provide for increased presence and engagement was not met for all participants. Still, some participants appeared to have had fun and play while using *Line-Storm*. Some of these participants likely experienced *Line-Storm* as art work, and so we would have found *preservers* for our work, who brought out its workly character, and who would belong to it just as we belong to it as its creators. This justifies our efforts.

The participants' prior knowledge is relevant when considering their responses to *Line-Storm*. *Line-Storm*, as a tool, exists not by itself but among a constellation of related tools; those related tools, some of which a given participant may be familiar with, and some of which they may not be familiar with, allow *Line-Storm* "to be this equipment that it is" [29]. A participant's degree of familiarity with related tools, such as an envelope, stamp, mail-box, and pencil and paper, help to determine what *Line-Storm* is for that participant. We see a nearly significant (r = 0.620, p = 0.056) correlation of current writing or drawing (by hand) practice to number of words written while using *Line-Storm*. We think that, this correlation indicates participants who regularly wrote or drew by hand were better able to experience *Line-Storm* as it was intended, and see its' authenticity.

## 7 CONCLUSIONS

We conceived our work, initially, as an entertainment system, to be used for one's own pleasure while writing in a journal. We followed that by hoping to jolt users out of complacent acquaintance with paper and pencil and present the writing tools and writing situation as if for the first time, to encourage the practice of writing and sending handwritten letters. We finished the work by attempting to enhance human creativity when working with a writing stylus and paper writing pad, by increasing participants' sense of presence and engagement. We found correlations and K-means clustering results that did suggest there was a group of participants who responded favorably to *Line-Storm*.

We expected that a direct approach to enhancing creativity may/would fail; we attempted to construct a system the use of which would be an end and not only a means [24], and hoped this might lead, indirectly, to enhancing creativity by encouraging play and playfulness. We provided a ludic environment for creative work, in which some users would focus on using the system, not expecting an outcome and will create their own play/outcome and accept what emerges or not–no quest, no winners, no points or gold to deliver outcome-based satisfaction. In a ludic system, therefore, the creative work (outcome is what it is) and the results would be a secondary consideration and may emerge by itself, an indirect result of the use of the system. We hoped participants in our experiments would find themselves "losing themselves," and a group of participants did tend to lose track of time while they used or performed with *Line-Storm*. We believe these participants became more absorbed while using the experimental system, exactly our intention. Losing oneself while using the system might open one up to creative energies, thoughts, feelings, and actions that would ordinarily not occur, as Nietzsche [51] wrote.

## 8 FUTURE WORK

Future work would include the following items, listed as follows: (a) Clean up Max/MSP code and put functionality back in place, that would allow the triggering of multiple thunderstorm samples in quick succession. We would reinstate the capability of triggering multiple thunderstorm samples in rapid sequence. A committee member responded positively to a version of our experimental system that did trigger multiple thunderstorm samples in rapid sequence, and their gratifyingly positive response to our system was what we had hoped to bring to our research participants. (b) Make a cover for the electronic components on the sensor-fob. This would provide better aesthetic appeal and would minimize distractions from things like blinking lights and hanging wires. (c) Extension to a mobile platform – a mobile platform would use a smartphone or tablet and would not require a laptop computer, which limits the places our system could be used. (d) Investigate the use of more miniaturized RF components. We do not need the relatively large antennae of the XBee radios, which can operate over a larger distance than we envision for the use of our system. Bluetooth would provide the necessary range. (e) Investigate using more miniaturized micro-controller boards. The Arduino Fio v3 was the smallest board we found, when we began our work, with all the functionality we needed. A smaller board would make a less intrusive sensor-fob. (f) Experiment with different styli, including a paintbrush, a child's crayon, a marker, a piece of chalk, a paint roller, and so on. Attaching a contact microphone to the surfaces used with many of these would probably produce a suitable-strength vibration for use with our system. (g) Experiment with a baton-type stylus like the one used by Paradiso and Machover in the Brain Opera. (h) Investigate a wrist-worn appliance to augment or replace the motion-tracking capability of the stylus sensor-fob. (i) Gather more data involving a larger sample size. (j) Vary the type of music listened to during the control condition. (k) Consider ways to run experiments without wearing out participants by making excessive demands on their attention. (l) Experiment with a multi-user system. Users could be situated in the same place or could communicate via a computer network such as the internet. (m) Collaborate with a composer of music or a composer of electroacoustic music. We discarded our attempts at constructing an interactive generator of electroacoustic music.

Collaborating with a person skilled in the creation of electronic music would be of great benefit in future as well. Specifically, such collaboration would improve the system by mapping accelerometer data to sonic parameters in an effective way, something we were unable to do to our satisfaction. More data points will also help our analysis; we would have liked to have run more experiments. We used statistical techniques that were designed for small samples, the Student's t Test and the Wilcoxon Signed-Rank Test. Our K-means clustering analysis would have been improved, we believe, by the addition of more data points.

Finally, it has occurred to us that augmentation itself is innovative. Augmenting means a possibility that is completely different than the original. The *Preservers of Line-Storm*, in our experiments, showed that there is promise for our augmented interface–it may not enhance the sense of presence and engagement and lead to more creative writing, which was the hypothesis we had hoped for, but, as we discussed, creativity is difficult to capture anyway. Still, our work provided a completely different experience through augmented interaction to creative writing which enhanced the user experience, which simple creative writing, using ordinary pen and paper, or word-processing software *cannot* provide.

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
