# OpenReview forum: "Line-Storm Ludic System: An Interactive Augmented Stylus and Writing Pad for Creative Soundscape"
_graphicsinterface.org/Graphics_Interface/2020/Conference — Submitted to GI 2020_

### Official Review · AnonReviewer1 · 2019-12-30
**Interesting system and study, but missing figures make understanding difficult and missing implementation details exclude possibility of replication**

**Confidence:** 4
**Rating:** 3

**Review:**

This paper describes a system for augmented writing and drawing, which the authors evaluated with a user study and which they also place in context of various philosophical writings. The outcome of the study seems to indicate that there are two distinct groups of users: those who perceived the system as just another way of writing/drawing, and those who engaged with the system as a creative tool.

Overall, this paper was somewhat of a challenge to review, as I believe there may have been a technical problem or incorrect document submission. The paper contains only one figure (at the top of page 1), though it does reference many others within the text. The paper contains one or two unrendered references (e.g., [FirstSecondAuthor] in section 3.2.1) and a mixture of citation styles (e.g., “(Springob, 2015)” vs “[34]” vs calling out works by name within the text as in “Taskscape is a complex term from Ingold’s ‘The Temporality of the Landscape’”). That said, most of the prose is easy to follow, if a bit repetitive and overly-detailed in some places. I have detailed comments and questions below about the study, system, and philosophical components of the paper, but unfortunately its readability makes it hard for me to recommend it for publication without substantial rewriting and a re-review with figures included.

Study: What was the prompt that was given to users? The authors mention that they removed data related to two users who used the system for drawing rather than writing, but I am not certain how to interpret that: were the users going against the prompt, or were they within the boundaries that were set by the experimenters? What was the “control condition” (presumably writing or drawing without LineStorm turned on)? Was the order of control and experimental condition randomized across participants to reduce ordering effects? What was the payoff for users who participated? It seems to be implied that extra credit was given to participants, although this isn’t explicitly stated. Were they still given extra credit if they chose not to continue with the study at any point? Did the authors track which users had more experience in creative writing/drawing with any kind of intake survey? In general, I found this section challenging to understand: I would encourage the authors to read it again with the idea that they want another researcher to be able to re-do their study in the same way. The data analysis from the study is also a bit hard to understand (due to the missing figures). I would like to see a bit more exposition on the k-means clustering, since the authors argue that is their primary result. Was the k-means clustering performed on all the answers to all the survey questions, or only a subset? What were the questions on the survey? Was there a particular question that separated the Preservers of Line-Storm most clearly from the other group of users, or were their results just slightly different on all questions?

System: The authors include a lot of technical details of parts and wiring for the Line-Storm system prototype. I believe that the physical components of the system would be easily replicable from their description (though I would encourage the use of a wiring diagram, e.g., from Fritzing https://fritzing.org/home/ to cut down on the needed exposition). The digital component, though, would be more challenging for me to replicate. The details of how the system reacts to user input are not included in the implementation section; it’s possible that they are in the unrendered [FirstSecondAuthor] paper? In an earlier section, the authors mention “The sounds made, while writing or drawing, are captured using a contact microphone and are played through headphones. Sounds of natural phenomena–the sounds of a thunderstorm–augment the writing or drawing experience.” I’m not clear what triggers the inclusion of additional sounds: is it related to accelerometer readings? When the authors later mention a comment from a committee member who suggested allowing rapid-fire triggerings of thunderstorm sounds, it suggests to me that there is also some kind of time-delay currently implemented in Line-Storm which limits sounds, though I am not certain how it works. Given MaxMSP’s graphical programming nature it could be appropriate to include a figure of the developed patch here, as well, to aid in understanding and replication, or alternatively a system flow diagram.

Philosophical underpinnings of Line-Storm: The authors clearly have a good grasp of the ludic literature to which their work relates, and walk the reader through some interesting references. As my background unfortunately does not include many of these, I found myself looking up a lot of terminology (e.g., “ thingly character” and “part of a matrix of all equipment”). It would be helpful to include a few definitions for these sorts of terms: the audience of this paper may be very diverse in their backgrounds, and likely not everyone who reads it will be familiar with Heidegger’s essay.

Overall: This paper presents an interesting system with some intriguing first results from a user study. However, because of missing figures, it is challenging to understand, and some sections would make replication difficult.

---

### Official Review · AnonReviewer2 · 2020-01-06
**Potentially incomplete submission with an unclear research question and results**

**Confidence:** 4
**Rating:** 3

**Review:**

This paper describes Line-Storm, a stylus-based system that augments the writing process with audio. The paper describes the philosophical underpinnings and inspiration behind the system, the design of the stylus itself, and reports on the results of a user study with line storm. The results did not find a significant difference between Line-Storm and a control condition but found that some participants were more responsive to the study and stylus than others.


I found this submission very difficult to review for a few reasons, and each of these reasons prevent me from recommending this paper for acceptance in its current form.


First, it appears that some pages of the submission may be missing or were accidentally omitted. In many places there are references to Figure 17, 21, 16, 22, etc. however there is only one figure in the paper, Figure 1 (first page). This makes it difficult to understand the relevance of some of details in the system description because the level of detail seems unnecessarily fine-grained, but the figure citations suggest that such details could be crucial to replicate the device in the future (e.g., does the reader really need to know that five wires were soldered between the PCBs to replicate the system?, most likely no but without the figures it is impossible to know). It also seems like a Section 3.3, which may have detailed the software side of Line-Storm, is missing. The paper doesn’t have any details on the actual sound augmentations that are performed and which user interactions cause them to be created / played / rendered.


Second, there is no description of the study methodology, i.e., equipment / apparatus, tasks, measures recorded and questionnaires administered, task order or counterbalancing, etc. (perhaps the rest of Section 4 was accidentally deleted?). Because of these omissions, it is impossible to understand or replicate the study that was conducted, determine if the statistics that were performed are correct, if the conclusions correctly follow from the results, or what the contribution of the study is (i.e., Section 5.3 and 6 cannot be properly reviewed without the missing information about the study). There is also confusion about the number of participants (13? 12? 10?) that needs to be fixed, and a power analysis to account for the small number of participants (if there were actually 10) is missing.


Thirdly, I found the Introduction and Section 2 much too focused on the philosophical notions of play, flow, and creativity. While I can appreciate that there was inspiration drawn from a great deal of scholarship and other’s writings, it overshadows and obfuscates the research questions this paper was attempting to explore. This is especially clear in 2.0.2 Previous and Related Work, where the reader does not get a clear picture of how this work extends, supplants, contrasts, or complements the HCI literature (it is unfortunately that of the 64 references, only 3 are from HCI venues (TEI, CHI, OzCHI)).


Lastly, there is one citation in the paper that suggests this paper may be an extension of prior work or possibly a concurrent submission, i.e., “More programming and other details can be found in [FirstSecondAuthor].” While I do respect the desire to maintain anonymity, providing this citation in this way is problematic because (i) those details are not included in this submission and (ii) reviewers cannot read this other citation to determine how similar (if at all) this paper and the citation are. It would have been beneficial to provide the actual citation and refer to the citation’s work in the third person (e.g., “We used the programming approach / method proposed by X et al. in [Y].”).


Other Notes:
-	Musc Grip is misspelled
-	Two different citation styles are used i.e., [X] and (X, XXXX).
-	Section 5.2 can be removed because it does not add anything to the submission and the description in 3.1 can do away with a description of MAX/MSP unless it is important for the reader to know such small details as the different types of inputs and outputs
-	Effect sizes are missing in Section 5, as are the test statistics and degrees of freedom
-	No details were provided about the k-means clustering “tests”
-	What is a “creative worker”?

---

### Official Review · AnonReviewer3 · 2020-01-07
**Incohesive paper with lots of irrelevant references, reads like a shortened version of a thesis with important parts like study methodology missing.**

**Confidence:** 3
**Rating:** 2

**Review:**

This paper describes the use of Line-Storm, an interactive system augmenting a mechanical pencil that produces sounds as the writer writes on a paper pad, to enhance creativity. No significance was found in doing so, but the authors stated that some participants wrote more in the study and reported greater presence and engagement, indicating a correlation.

I had a hard time understanding this paper. It jumps between creativity, popular believes, and performance, with little cohesiveness. I don’t see any particular reason to read through arguments of augmentation or replacement — just state that you want to augment a common, familiar tool to improve the creative process is sufficient. Similarly, the comparison between Line-Storm and puzzle (Section 6.5) as well as other art work (Section 6.6) adds very little to the discussion of creativity, and feel very out of place. I would argue rejecting this paper.

Pros
-Draws an interesting connection between creativity and presence and engagement.
-Quite detailed description of the sensor-fob construction, except all the figures are missing.

Cons
-Inconsistent reference style (e.g., Steven Jesse Bernstein 1992, Csikszentmihalyi 2014).
-Related work has a lot of random concepts (e.g., play, gamification, privacy) and unnecessary quotes (e.g., Tod Machover, Heidegger). It is also packed. Consider breaking it into similar groups like Music-augment objects, Handwriting & learning, …etc., and pick those that are most relevant.
-Figure references in Section 3.2 are nowhere to be found, making it almost impossible to visualize how the sensor-fob looks like.
-Description of the study methodology is missing. What did the participants do during the study? Information about dependent variables (sense of presence and engagement) are only briefly mentioned in the summary of results. It is also unclear whether the study was a within- or between-participant study.
-Section 6.3 is highly speculative and it is unclear why “attention” is discussed there.
-Section 6.4 is very abstract and I find many of the descriptions superficial — it is really just adding some sound to a writing tool without even hiding the electronics to make it more aesthetically pleasing or non-intrusive.

Minor things
-Where were the two sites the study took place? Laboratory? Classrooms?
-The one-sentence first paragraph for Sections 3-6 reads a bit odd and obvious. Typically it has more content to highlight the main points of the section.

It is quite apparent that this paper is a trimmed down version of a longer thesis of sorts. As indicated by words like “committee”, incorrect figure references (e.g., Figure “17”), inconsistent/missing references (e.g., [FirstSecondAuthor]), and justification of SPSS (most researchers in this community know what it is and have used it).

The authors might be able to better articulate their intention by reporting comments/answers made by the participants, particularly the “preservers”. Currently	the only findings are abstracted to significances and correlations, which are not that helpful.

Lastly, I find it disturbing to include LSD as a way to enhance creativity especially when this work is unrelated to drug use.

---

### Meta-Review · Area_Chair1 · 2020-01-09

**Recommendation:** Reject
**Confidence:** 5

**Metareview:**

All three reviewers unfortunately felt that this paper was not above the bar in its current state; the paper is currently hard to review because it seems incomplete (i.e., missing figures / sections) and much of the related work / introduction concepts don't seem related to the stylus design, motivation, or study. In the current form, it is unclear what the digital aspect of the system does (i.e., which sounds are generated and when) and difficult to evaluate the results due to missing details about the study methodology.

---

### Decision · Program_Chairs · 2020-01-11

Reject